# Assessing Kidney Graft Viability and Its Cells Metabolism during Machine Perfusion

**DOI:** 10.3390/ijms22031121

**Published:** 2021-01-23

**Authors:** Maria Irene Bellini, Francesco Tortorici, Maria Ida Amabile, Vito D’Andrea

**Affiliations:** 1Azienda Ospedaliera San Camillo Forlanini Hospital, 00152 Roma, Italy; 2Department of Surgical Sciences, Sapienza University, 00152 Rome, Italy; mariaida.amabile@uniroma1.it (M.I.A.); vito.dandrea@uniroma1.it (V.D.); 3National Nuclear Physics Institute, INFN, 95123 Catania, Italy; francesco.tortorici@ct.infn.it; 4Department of Physics, Catania University, 95123 Catania, Italy

**Keywords:** ischemic reperfusion injury, kidney transplantation, organ preservation

## Abstract

Kidney transplantation is the golden treatment for end-stage renal disease. Static cold storage is currently considered the standard method of preservation, but dynamic techniques, such as machine perfusion (MP), have been shown to improve graft function, especially in kidneys donated by extended criteria donors and donation after circulatory death. With poor organ quality being a major reason for kidneys not being transplanted, an accurate, objective and reliable quality assessment during preservation could add value and support to clinicians’ decisions. MPs are emerging technologies with the potential to assess kidney graft viability and quality, both in the hypothermic and normothermic scenarios. The aim of this review is to summarize current tools for graft viability assessment using MP prior to implantation in relation to the ischemic damage.

## 1. Introduction

Kidney transplantation represents the golden treatment for end stage renal disease (ESRD) with longer life expectancy and superior quality of life, compared to the current alternatives of renal replacement therapies. However, one of the major limitations is the lack of suitable organ donors, resulting in waiting list mortality [1].

To expand the pool of suitable organs, the transplant community has progressively opened up to the use of donors after circulatory death (DCD) in which a variable time of warm ischemia (WIT), i.e., a period of low or no oxygenation and blood perfusion at body temperature affects the organ before the cooling down during organ retrieval [2,3]. Another organ source increasingly being used are expanded criteria donors (ECDs), defined as any donor aged ≥60 years or over 50 years with ≥2 of the following conditions: hypertension, terminal serum creatinine equal or greater than 1.5 mg/dL or death resulting from an intracranial hemorrhage [4].

Although a potential solution to organ shortage, a significant proportion of these other than standard kidneys develop delayed graft function (DGF) or, if the damage is more sustained, primary non function (PNF) after transplantation, with resultant significant morbidity and mortality risks for the recipients [5].

The aim of this review is to provide an insight on the reasons for damage in the acute phase of clinical ischemia reperfusion injury, understanding the mechanisms that are driving these processes in the clinical scenario. This will allow a preemptive assessment, potentially beneficial to predict kidney function and the selection of the possible transplant candidate [6], in consideration of an appropriate donor-recipient match and increase the organ utilization rate [7].

## 2. Pathogenesis of the Ischemic Injury

Since the beginning of clinical transplantation, the ischemic period consequent to the lack of oxygen and blood supply after the organ retrieval, has been considered a major limiting factor affecting short- and long-term outcomes [8]. Thus, the initial approach from Joseph Murray, the performer of the first successful kidney transplant, was to shorten to the minimum the ischemic period, operating in two adjacent theatres. In this way, once the organ was retrieved, it was immediately implanted into the recipient’s body, the donor’s twin [9].

To elucidate the organ damage and the possible viability impairment resulting from the ischemic period, it is important to look at the pathophysiology of the ischemic-reperfusion injury (IRI).

Cell metabolism is directly related to the oxygen and blood supply the vital body delivers: the lack of perfusion consequent to the donor death and the retrieval process alters therefore the electron transport chain within the mitochondria, the energy central of the cell (Figure 1). 

Inevitably, there is a decrease of the adenosine-triphosphate (ATP) production, with a dysfunction of the sodium-potassium and calcium pumps that are directly ATP dependents. The failure of Na^+^-K^+^-ATPase pumps causes retention of sodium within the cells, decreasing the activity of sodium–hydrogen exchanger pumps (Na^+^-H^+^ pumps). In the same way, calcium pumps (Ca^2+^-ATPase pumps) on the endoplasmic reticulum also become dysfunctional, limiting calcium reuptake. In cells, the accumulation of hydrogen, sodium and calcium ions causes hyperosmolarity, which leads to water flow into the cytoplasm and cell swelling, with organ oedema. The retention of hydrogen decreases cellular pH, leading to impaired enzyme activity and clumping of nuclear chromatin, with DNA damage, acidosis and altered protein synthesis [10].

This evidence of metabolic dysfunction as driver of the IRI implies a critical role played by ATP during organ preservation. The ischemic state induces anaerobic metabolism, leading to a lower level of ATP production and failure of ion-exchange channels, in parallel to succinate accumulation [11]. These changes are accompanied by opening of the mitochondrial permeability transition (MPT) pore, which dissipates mitochondrial membrane potential and further impairs ATP production [12].

Since ATP is the energy currency of cells, impaired viability and damage will result in ATP depletion. This can be indirectly measured from the increased concentrations of extracellular hypoxanthine, a central intermediate in the metabolism of ATP; thus, cell damage secondary to ATP depletion can also be measured from extracellular hypoxanthine, that can easily cross cells membranes.

In consideration of the above, slowing cell metabolism during hypothermic conditions, could save cellular damage and lethal consequences. By reducing the temperature and cellular metabolism, the use of oxygen and the rate of depletion in energy substrates, such as ATP, will reduce in parallel. At hypothermic temperature (4 °C), the rate of metabolism is 10% of that of normal physiological temperature and the rate of chemical reactions of interest is 40% of that in organs perfused at body temperature (37 °C), according to the van ’t Hoff equation. In addition, the Arrhenius relation states that as temperature decreases, the thermal excitation of molecules also reduces and their chemical interactions [12].

Other biochemical events occur during ischemia that do not contribute to the ischemic injury immediately, but that later activate a cascade of toxic events at the time of reoxygenation with blood reperfusion, and in this way exacerbating the previous tissue injury.

During reperfusion, the restoration of blood supply determines a modification of the oxygen structure, with the captation of a single electron; this produces the “superoxideanion”, the first reactive oxygen species (ROS). When ROS production increases, oxidative damage follows in parallel and if the cell’s mechanisms of repair are insufficient, i.e., antioxidant systems to scavenge the reactive intermediates lack or are not reimplemented, the consequent lesions evolve into glomerular sclerosis, determining irreversible loss of function, provided that the cell survives.

Among the possible ROS originators, mainly oxidoreductases, particular attention should be reserved to mitochondrial nicotinamide adenine dinucleotide hydride (NADH) and nicotinamide adenine dinucleotide phosphate (NADPH) oxidase. Interestingly, unlike other enzymes that produce ROS secondary to their specific catalytic process, NADPH oxidase is in fact the only enzyme whose primary function is to produce ROS [13]. Furthermore, particularly for the kidney, the primary origin of ROS in the renal cortex is the NADPH oxidase itself [14,15].

More in details, NADPH donates hydrogens to glutathione and thioredoxins, which in turn are used by glutaredoxins, peroxiredoxins and glutathione peroxidases to neutralize ROS, with the acquisition of the reductive potential. Thus, NADPH could be considered as the ultimate donor of reductive power to the antioxidant systems and scavengers of the toxic reactive intermediates accumulation.

NADPH can be generated by numerous ways, including the cycle catalyzed by the malic enzymes, isocitrate dehydrogenases and folate dehydrogenases; yet the principal source are two enzymes of the oxidative branch of the pentose phosphate pathway (PPP): the 6-phosphogluconate dehydrogenase (6PG) and the PPP rate-limiting enzyme glucose-6-phosphate dehydrogenase (G6PD) [16].

## 3. Consequences of Ischemic Reperfusion Injury

The clinical manifestations of ischemia-reperfusion injury are diverse. Pathologically, ischemic acute kidney injury (AKI) is characterized by sublethal and lethal damages in renal tubules, especially in the proximal tubules. Podocytes, the highly differentiated glomerular epithelial cells lying in the outmost layer of the glomerular filtration barrier, seem to be the more susceptible to the damage. The interdigitations of podocytes intertwin with those of the surrounding cells to constitute the slit diaphragm, which represents the ultimate protection to avoid excessive protein loss from the glomerular filtrate [17].

As above mentioned, the oxidative stress could cause cell death; this is in fact shown in renal tubules of AKI patients’ kidneys; if the damage is sustained but not lethal, renal tubules possess the capacity to repair and, according to the own reservoir capacity, mainly related to the quality of the kidney itself, i.e., living donor versus ECD or DCD, or young versus aged donors, the functioning parenchyma is replaced by fibrotic tissue, with progression towards chronic deficiency [18,19] and loss of filtration capacity.

### 3.1. Delayed Graft Function

Renal graft function after transplantation is typically defined as immediate, delayed (DGF) or primary non function (PNF). The majority of centers define DGF as the requirement for dialysis within the first week after transplantation.

The diagnosis is based on low urine output, slow decline in serum creatinine levels and increased metabolic instability. The definition of functional DGF (f-DGF) consists with the absence of a decrease in serum creatinine levels of at least 10% each day for three consecutive days within the first week post-transplant, but not including patients in whom acute rejection or calcineurin inhibitor toxicity is proven on biopsy [20].

The damage to the podocytes in the tubules, also known as acute tubular necrosis (ATN), mainly caused by the IRI, is considered the main cause of DGF after transplantation [21]. DGF has an influence on both short- and long-term outcomes, increasing the risk of acute rejection (AR), parenchymal scarring and reduced graft function and survival. It also has a significant hospitalization cost, with a higher overall recipient morbidity [6]. Rates of DGF typically affect one third of the deceased donor kidney transplants [22] and are dependent from the intrinsic quality of the organ, with a general less incidence for high quality organs as for example those retrieved from living donors [23].

### 3.2. Acute Rejection

The effects of organ ischemia, namely loss of ATP and increased ROS production secondary to the anaerobic metabolism, lead to an accumulation of lactic acid, malfunction of the Na^+^/K^+^ ATPases and oxidative damage. Furthermore, the stress deriving from the blood reperfusion of the injured tissue could paradoxically exacerbate the generation of ROS and the total amount of intracellular damage, therefore IRI may increase the rate of the inflammatory response consequent to the cells death releasing chemokines and other toxic molecules. This intracellular release is targeted as a threat to the organism by the receptors of the innate immune system that activate inflammatory cells and mediators. Thus, organs with a sustained ischemic damage are also attacked by the immune system with a synergic negative effect on graft recovery. There is evidence in fact that the inflammation generated as a consequence of IRI damage may worsen the function recovery and may stimulate dendritic cells to become mature, intercepting the graft tissue antigen and migrating to the lymphatic system. In this situation, the usual path generally involves presentation of the antigen to T cells, with the activation of the adaptive immune system and a sustained immune reactivity, known as AR against the transplanted kidney, both in the forms of humoral and/or cellular components [24].

## 4. Preservation Techniques

### 4.1. Static Cold Storage

Organ preservation has relied on cold static storage (SCS) to minimize the damage outside the body, since the retrieval and until transplantation, when the revascularization with the recipient’s blood brings the organ back to the normal metabolic conditions [8]. This preservation modality has represented the most common used method worldwide because of its simplicity and relatively low associated cost [25]. Practically, at the retrieval site, the kidney is flushed with cold preservation solution to remove the blood and be cooled down; then, it is stored in solution surrounded by crushed ice.

There are several preservation solutions available in commerce, all with the same common principle to limit and potentially prevent the described above tissue damage secondary to the ischemic injury. The basic formula is the presence of an impermeant to counteract edema and provide stability to the cell skeleton as well as a buffer with a balanced electrolyte composition to prevent the accumulation of intracellular acidosis and further minimize cellular swelling. The concentrations of sodium and potassium is variable and according to which electrolyte level is higher, they are classified as extracellular or intracellular, respectively [26].

Cold preservation is based on the principle that cooling an organ inhibits the enzymatic processes and the detrimental effects of the anaerobic phase; by reducing the temperature of 10 °C, there is a 2–3-fold decrease in cell metabolism happening in parallel, leading to a slower depletion of ATP [27] and therefore letting the cell survive a longer period outside the human body. Organ ATP stores are depleted by cold storage, and despite the fact that hypothermia abrogates some of the harmful effects of ATP depletion [14], graft endothelial damage and inflammatory responses are enhanced, the degree of which is related to the length of cold storage time [9].

### 4.2. Dynamic Preservation

The concept of dynamic preservation lies on the mechanism of an active perfusion of the organ versus the static preservation solution in storage with crushed ice. According to the temperature setting, we could mainly distinguish two different scenarios: hypothermic or normothermic machine perfusion. They are increasingly being adopted, particularly for organs retrieved from DCDs and ECDs. A recent meta-analysis systematically reviewed the evidence for viability and incidence of reperfusion injury in kidneys preserved with MP versus SCS [25], showing evidence of improved outcomes versus SCS, namely reduced DGF and PNF and increased 1 year graft survival.

The aim of dynamic preservation with MP is to facilitate restoration of cellular metabolism, on the contrary impaired by SCS. Furthermore, there is the possibility to directly modify the perfusion fluid with the delivery of organ directed reconditioning therapies. In this unique scenario, by establishing an isolated ex vivo platform with a metabolically active organ, therapies targeting ischemia reperfusion injury in real time could be delivered directly to the organ and limit systemic recipient exposure. In this regard, aside from the oxygen implementation [28], other drugs as prostaglandin, antibiotics, bicarbonate and heparin have been described, as well as mesenchymal cells in a recent pre-clinical trial [29].

#### 4.2.1. Hypothermic Machine Perfusion

Hypothermic Machine Perfusion (HMP) is based on a continuous flow of preservative solution that is recirculated within a sterile circuit at hypothermic temperatures, i.e., between 4 and 7 °C [6]. The perfusion pumped directly into the kidney favors a complete washout of blood and clots, improving in this way the penetration of the perfusion solution components in the parenchyma. Additionally, the flow dynamics allow the possibility of real-time viability assessment, and the potential provision of substrates to sustain the metabolic activity, such as pharmacologic agents or nutrients.

The active pump of preservation solution is contraposed to the static nature of cold storage. This also sustains a more active intervention of the clinician in monitoring the organ damage during the ischemic phase and eventually intervene to recondition the injury, if recognized [8]. As for the case of SCS, HMP slows down the cell metabolism, reducing oxygen requirements and ATP depletion. Additional research focuses on a vasoprotective effect of HMP, by potentially maintaining the hemodynamic stimulation of the endothelium, decreasing the vasospasm, facilitating the expression of protective flow dependent genes and maintaining the patency of the vascular bed [30,31,32]. In this respect, the use of pulsatile flow in HMP appears to be the important factor in HMP effects versus non-pulsatile flow, in order to ensure benefits of HMP compared to SCS [6]. Perfusion dynamics parameters related to perfusate flow through the kidney and commonly used in the clinical practice are pulse pump rate, perfusate temperature, perfusion pressure (systolic, diastolic and mean pressures), perfusion flow index (PFI) and intraparenchymal vascular resistance.

The evidence examining perfusate analysis from HMP kidneys compared to flush solution from SCS stored kidneys demonstrated that, after HMP, the kidneys have significantly reduced proinflammatory cytokine expression compared with SCS controls, providing a potential mechanism for HMP to permit a reduction in leukocyte activation and a decrease in IRI activated inflammation during reperfusion [33]. Yet, it remains to be ascertained the overall effect of HMP, as there might be additional components to be investigated for further potential benefit. For example, a recent randomized controlled trial [28] has demonstrated that active oxygen addition into the perfusate leads to a beneficial effect over HMP only, in kidneys retrieved from donors aged 50 years or older or after circulatory death. The improvement shown in the 12-month eGFR in transplanted kidneys undergone oxygenation during HMP, paths the way towards real chances of reconditioning before transplantation, actively contrasting the ischemic damage as a result of the anaerobic cell metabolism.

In addition to the capability of monitoring perfusion dynamics, the circulating perfusate can be sampled for levels of damage and injury biomarkers. In both instances, perfusion dynamics and biochemical perfusate analysis have been described in the assessment of organ viability and suitability for organ transplantation [34].

#### 4.2.2. Normothermic Machine Perfusion

Normothermic Machine Perfusion (NMP) aims to maintain the organ under a physiological temperature setting, to allow the continuation of the biochemical processes inherent to cellular metabolism, outside the human body. Ex vivo NMP is to be distinguished from in situ normothermic regional perfusion, consisting of the use of extracorporeal membrane oxygenation in donors after circulatory death, but with the organs still within the donor body.

Continuous perfusion of the kidney at warmer temperatures (34–37 °C) with the delivery of nutrients and oxygen has the advantage of avoiding the hypothermic injury and hypoxia, thus NMP seems to establish a more physiological environment while preserving the kidney. In addition, it may also aid recovery and prevent further injury occurring before the reperfusion with human blood [35]. As for the case of HMP, the dynamic preservation with NMP directly shows its superiority compared to SCS on both clinical and experimental settings [36,37].

To reestablish the completeness of cell metabolism during preservation and before the graft is actually transplanted and perfused by the recipient’s blood, it is necessary to provide the organ with nutrients and oxygen, thus an oxygen carrier, usually red blood cells, is required. There is growing evidence also of acellular perfusates, such as those making use of hemoglobin-based oxygen carriers, that could represent a cost-effective alternative [38].

At normothermia and in the presence of oxygen, cellular metabolism resumes, implying a higher likelihood of assessing both organ injury as well as residual function [27]. For instance, the overall macroscopic aspect, as well as the perfusion flow and the urine production constitute an increasingly utilized score to predict graft functionality after transplantation [36].

At current, most of the available evidence investigates normothermic kidney perfusion during a brief period (usually 1 h) before transplantation, due to the continue need to recharge with nutrients and additives the solution and eventually replace the excretion products of the metabolism of the cells. There is also description of human kidneys, not deemed transplantable, that have been normothermically perfused for 24 h, via a urine recirculation circuit [39].

Practically after retrieval, the kidney is flushed with cold perfusion solution, and either immediately perfused on a portable machine perfusion device, or transported using transiently SCS back to the recipient hospital for perfusion on site. Ex vivo NMP was commenced [40] via a pediatric cardiopulmonary bypass and membrane oxygenator to provide the kidney with oxygenated red blood cells suspended in crystalloid at 37 °C. As for the HMP case, perfusion dynamics parameters allow viability assessment and the macroscopic appearance of perfusion and the urinary production provide information regarding the parenchymal functional status [41]. In addition to the capability of monitoring perfusion dynamics, the circulating perfusate (or urine) can be sampled for levels of damage and injury biomarkers.

## 5. Viability Assessment via Machine Perfusion

With the progressively increase of comorbidities (diabetes, metabolic syndrome, coronary heart disease) affecting the waiting list candidates [42] and their overall associated risks to develop post-operative complications, along with the ageing of the ESRD population, the detrimental outcomes of the morbidity associated with the poor functionality of the implanted graft could have different risk acceptance thresholds for different recipients [43]. In other words, knowing the risk of developing poor graft function in real time during the preservation process, with the possibility also to measure the damage entity, as well as its possibility to recover with or without reconditioning, would provide an additional objective information for selecting a particular recipient for a particular kidney and thus pursuing a better donor-recipient match in the attempt to keep expanding the organ donor pool. Table 1 represents a review of the most broadly used methods for viability assessments of the kidney during preservation via MP (from the newest to the older up to 2000).

A list of the evidence in available tools is reported below:(1)During NMP macroscopic appearance of the perfused graft is available: the quality assessment score (QAS), is based on macroscopic appearance, mean renal blood flow and total urine output [41]. Kidneys are graded 1–5, with 1–3 scores considered suitable for transplantation: score 1 indicates the least injury and 5 the most severe. More in details, the score is built up by a combination of the perfusion assessment parameters within 60 min from the start: grade I, excellent perfusion or global pink appearance; grade II, moderate perfusion with patchy pink/purple appearance which either remains or improves during NMP; grade III, poor perfusion, consisting of global mottling and purple/black appearance constantly throughout NMP. In addition, thresholds of renal blood flow (<50 mL per min per 100 g) and total urine output (<43 mL per min per 100 g) gives additional single points each to be combined with the macroscopic grades (I-III) for the final assessment score.(2)Pressure, flow and resistance readings measured during MP are used as viability assessors, although they cannot be considered as stand-alone criteria, since their relative predictive value is low. The rationale for the use of perfusion parameters stands on the structure of the renal vascular system itself, very rich in capillary network with filtration function [77]. The release of vasoconstrictors from this capillary network (single one-layer endothelium) following the ischemic and inflammatory insults, determines accumulation of erythrocytes and microthrombosis, eventually leading to a diminished flow and increased resistance in the graft [26]. Furthermore, the hypoxia is directly responsible for endothelium cell activation, synergically favoring a pro-coagulant and pro-inflammatory phenotype of the renal vasculature, with consequent disruption of the blood flow, and increased leukocyte infiltration, with a further decline in kidney function. On this basis, increased renal vascular resistance and low intraparenchymal flow are expressions of tissue damage.(3)Glucose consumption: the difference between the concentration into the arterial inflow and venous outflow could estimate the aerobic respiration and energy activity of the kidney cells. Several ways to measure glucose consumption have been described, including metabolic profiles via noninvasive MR spectroscopy [78]. The rationale lies on the estimation of cells viability in view of their metabolic utilization of carbohydrate energy sources, as it physiologically occurs when the organ is within the human body. The pattern of shutting down metabolically is peculiar of the kidney suffering from oxidative stress and shifting towards anaerobic energy production, while renal perfusion decreases.(4)Oxygen consumption: the blood concentration of oxygen is measured to indirectly assess mitochondrial activity: there is a linear relationship between Na^+^ reabsorption and oxygen consumption, in fact Na^+^ reabsorption is mediated by an energy-dependent (Na^+^/K^+^ ATP-pump) process [12]. Recent studies have shown that oxygen administration during HMP increases oxygen consumption from the cells and improves kidney function (GFR) in the transplanted kidneys [28]. There are a variety of formulas currently used that differ on the parameters to be considered [55]. On a separate note, it is of relevance to estimate the calculation according to the temperature range, as in fact previously mentioned, cell metabolism is slowed down by the reduced temperature, therefore the oxygen requirement at hypothermic conditions if different from that at body temperature; furthermore, oxygen consumption during NMP is dependent on the oxygen concentrations offered to the kidney itself [79].(5)Measurements of final glycolysis products. A lack of oxygen causes accumulation of peculiar metabolites [80]: succinate/pyruvate, NADH, lactate (Figure 1). The measurement of tissue damage and estimated anaerobic metabolism is a major feature within ischemic organs, with correlation to the extent of warm ischemia time, as for example in the case of DCDs.(6)Measurement of ATP depletion or *ATP**/**ADP ratio*, as the key feature to determine if cell metabolism is predominantly oxidative or glycolytic. With the Na^+^/K^+^ ATPases block, the influx of free Ca^2+^ into the cells and the activation of phospholipases are direct consequences of the fall of ATP production [12]. Another indirect effect is also the increase concentration of transition metals as free iron, since its binding into the carrier proteins (transferrin, ferritin) is inhibited, too by the energy depletion. In this situation, there is also activation of the oxygen free radicals cascade, generating a vicious cycle in which the production of Nitric Oxide (NO), another commonly used measurement of cell viability, increases too [81]. NO has also a direct effect on vasoconstriction, thus relating to perfusion dynamics.(7)Viability of the kidney during machine perfusion can also be measured by sampling the perfusate for biomarkers of cellular injury [82]. In the hypothermic setting, the most commonly used are glutathione S-transferase (GST), as total-GST (t-GST) or its isoforms (alpha-GST and pi-GST), fatty acid binding protein (FABP), lactate dehydrogenase (LDH) and lactate levels. In the normothermic scenario, the most utilized are neutrophil gelatinase–associated lipocalin (NGAL) and endothelin-1 [39,83].(8)Microdialysis: a tissue sampling technique using a small (normally 600 μm diameter) probe with a semipermeable membrane at the tip. The inside of the membrane is perfused to maintain concentration gradient across the membrane between the extracellular fluid and the probe. This creates a dialysate stream specular to the tissue concentrations of analytes, as for example glucose and lactate. There is evidence in literature of real time in vivo monitoring, demonstrating that using online microdialysis can provide information on the metabolic state of organs during preservation [84].(9)mRNA profiling: defective postreperfusion metabolic recovery directly associates with incident delayed graft function and there is evidence of some ischemia induced omics that could be used as predictors of tissue injury [85]. Specific mRNA expression of several glycolytic and gluconeogenic enzymes could evaluate renal glucose metabolism or the degree of inflammation and cytokine production, secondary to the ischemic insult.(10)Flavin mononucleotide (FMN) levels in the acellular perfusate after 30 min of hypothermic perfusion, as a result of damaged mitochondria releasing their content into cytoplasm [44]. Physiologically, FMN is non-covalently bound to a subunit of the mitochondrial complex I and its dissociation with release at a cytoplasmatic level is an effect of the ischemic injury, where the MPT is damaged, with ROS production and increased toxicity [86].

## 6. Conclusions

The development of dynamic organ perfusion technology has significantly increased the possibility to assess the parenchymal cells metabolism during preservation, thus offering an additional tool to consider organ viability, particularly for those organs retrieved from broader acceptance criteria, i.e., ECDs and DCDs. In this way, there is evidence of a concrete chance to expand the organ pool, through the variety of MP approaches, by tailoring each preservation parameter (temperature, oxygen and/or nutrient delivery, location) to each different organ, on the basis of the presumed ischemic injury.

The concept of dynamic preservation itself is already beneficial to the organ metabolism, directly affecting the renal microvasculature and thus slowing down the resulting ischemic vasoconstriction. This could be estimated via the perfusion parameters of increased renal resistance and impaired intraparenchymal flow. Additionally, several indirect methods, as oxygen and glucose consumption, ATP depletion/production, lactate and biomarkers concentration could provide an insight of what is the current organ energy status to further drive reconditioning interventions. Yet, to date, there is a lack of the complete understanding of the mechanisms regulating the ischemic cell damage, thus a full viability assessment during dynamic organ perfusion is not available.

In summary, the more information related to viability assessment during organ preservation is provided, as in the case of MP compared to SCS, the highest is the accuracy in predicting the future function of a given graft. Further studies are warranted to reduce the inappropriate organ offer decline, possibly integrating a multidisciplinary approach to complement clinical or omics data with the variables examined via MP technology.

## Figures and Tables

**Figure 1 ijms-22-01121-f001:**
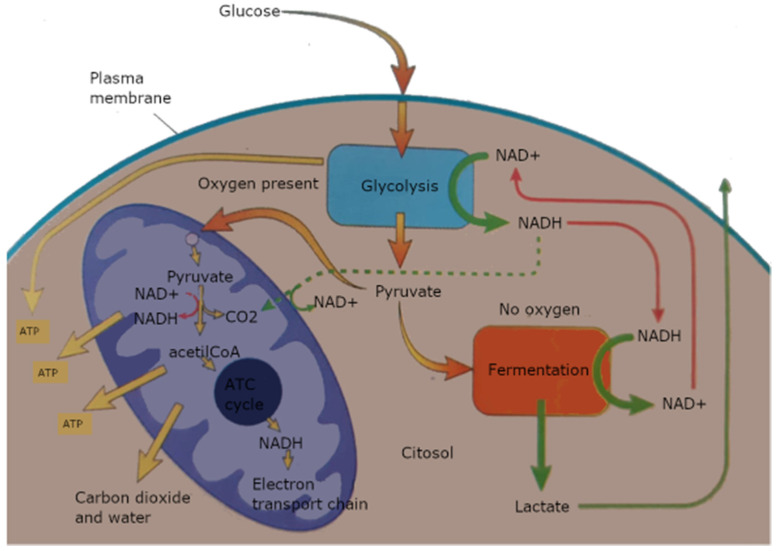
Cell metabolism. Glycolysis produces pyruvate and NADH in the cytoplasm. In the absence of oxygen, pyruvate is reduced to lactate (or to another fermentation product as ethanol), that is eliminated and NAD+ is reutilized to continue glycolysis. In the presence of oxygen, pyruvate is decarboxylised and linked to A coenzyme (CoA), to produce NADH. NADH donates its electrons to mitochondria, until the final chain element, that is the oxygen. The energy released by NADH during electron transports is used to produce ATP.

**Table 1 ijms-22-01121-t001:** Viability assessment parameters during ex situ Machine Perfusion. Legend: AAP: alanine aminopeptidase; ATP: Adenosine triphosphate; DBD: donor after brain death; COR: controlled oxygenated rewarming; DCD: donor after cardiac death; ECD: extended criteria donor; FABP: fatty acid binding protein; FMN: Flavin mononucleotide; GFR: glomerular filtration rate; GST: glutathione S-transferase; HMP: hypothermic machine perfusion; H-FABP: heart-type fatty acid binding protein; IL: interleukin; LPOP: lipid peroxidation products; LDH: Lactate dehydrogenase; KIM-1: kidney injury molecule-1; MDA: malondialdehyde; mRNA: messenger RNA; miRNA: microRNA; NAD: Nicotinamide Adenine Dinucleotide; NAG: N-acetylβ-D-glycosaminidase; NGAL: neutrophil gelatinase–associated lipocalin; NMP: normothermic machine perfusion; SNM: subnormothermic machine perfusion; TBARS: thiobarbituric acid reactive substances.

Author	Donor Type	Perfusion Type	Viability Assessment	Model
Wang et al. [44]	DCD/ECD	NMP	Macroscopic appearance, mean renal blood flow and total urine output; FMN in the perfusate	Human
Gomez-Dos-Santos et al. [45]	ECD	HMP	miRNA in the perfusate	Human
Bellini et al. [6]	SCD/DCD	HMP	Perfusion dynamics	Human
Weissenbacher et al. [39]	DBD/DCD	NMP	Perfusion parameters; NGAL and KIM-1 levels in the perfusate; pO2 and pCO2 levels; glucose measurement; lactate levels; urine production and sodium levels in perfusate and urine	Human
Juriasingani et al. [46]	DCD	SNM	Fluorescent marker that binds to double-stranded DNA	Animal (pig)
Gregorini et al. [47]	DCD	–	Lactate, LDH, MDA, glucose and pyruvate in perfusate samples, RNA in the perfusate	Animal (rat)
Van Smaalen et al. [48]	DCD	HMP	Extracellular histone (H3) in perfusate samples	Human
Hamaoui et al. [49]	DCD	HMP	Creatinine clearance, oxygen, glucose consumption, lactate, microdialysis	Animal (pig)
Sevinc et al. [50]	DCD	HMP	Perfusion parameters; GST levels in the perfusate	Human
Hosgood et al. [41]	Kidneys retrieved, but not implanted	NMP	Macroscopic perfusion, perfusion parameters, urine output	Human
Schopp et al. [51]	SCD	COR	Oxygen consumption, total content of NAD, functional activity of caspase 9 in mitochondria	Animal (pig)
Guy et al. [52]	SCD/DCD/ECD	HMP	Perfusion parameters; metabolomic profile via nuclear magnetic resonance: glucose, inosine, leucine and gluconate concentrations.	Human
Gomez et al. [53]	ECD	HMP	Perfusion parameters	Human
Buchs et al. [54]	DCD	HMP	ATP levels via Magnetic Resonance Imaging	Animal (pig)
Bunegin et al. [55]	ECD	HMP	Perfusion parameters; oxygen consumption	Human
De Vries E. et al. [56]	DCD	HMP	Renovasculature circulating volume	Human
Gallinat et al. [57]	SCD	HMP	Perfusion parameters, oxygen consumption, urine production, clearance of creatinine	Animal (pig)
Nagelsch et al. [58]	ECD	HMP	GST, LPOP, lactate and LDH levels in the perfusate	Human
Hoogland et al. [59]	DCD	HMP	GST, LDH, H-FABP, redox-active iron, IL-18, and NGAL in the perfusate	Human
Wilson et al. [60]	DCD	HMP	Perfusion parameters; GST in perfusate samples and mitochondrial electron microscopy	Animal (rat)
Jochmans et al. [61]	SCD, ECD and DCD	HMP	Perfusion parameters	Human
Tolstykh et al. [62]	SCD	HMP/NMP	Oxygen consumption, potassium-hydrogen gradient, perfusion parameters, GFR, fluorescence to investigate cell membrane viability	Animal (rat and dog)
Weegman et al. [63]	DCD	HMP	Oxygen consumption	Animal (pig)
Matsuno et al. [64]	DCD	HMP	Perfusion parameters	Human
Koetting et al. [65]	SCD	HMP	Oxygen consumption, LDH in perfusate, creatinine and urea concentrations, functional activity of caspase 3	Animal (pig)
Navarro et al. [66]	DCD	HMP	Perfusion parameters	Human
Bagul et al. [67]	DCD	HMP/NMP	Perfusion parameters, oxygen consumption, ATP levels, Von Willebrand factor	Animal (pig)
Maathius et al. [68]	DCD	HMP	Perfusion parameters, TBARS, NAG and AAP activity in urine, microcirculation, mRNA, histology	Animal (pig)
Wilson et al. [69]	DCD	HMP	Perfusion parameters, perfusate enzyme viability assay (GST), perfusate pH and lactate concentrations.	Human
Baicu et al. [70]	DCD	HMP	Kidney weight, perfusion parameters, glutamate and ammonium in the perfusate	Animal (pig)
De Vries B. et al. [71]	DCD	HMP	Redox-active iron, LDH, GST and hemoglobin concentrations	Human
Minor et al. [72]	DCD	HMP	Electron microscopy (vascular endothelium), ATP levels in tissue homogenates	Animal (pig)
Gok et al. [73]	DCD	HMP	Perfusion parameters; biomarkers of tubular injury: GST, AAP and FABP levels.	Human
Brook et al. [74]	DCD	HMP	Perfusion parameters,	Animal (pig)
Balupuri et al. [75]	DCD	HMP	GST levels in the perfusate	Human
Polak et al. [76]	SCD	HMP	Perfusion parameters, LDH, lactate level, pH and samples of perfusate for α- and π-GST.	Human

## Data Availability

The data supporting this review have been provided throughout the text.

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
