# Peer review of "Assessing Kidney Graft Viability and Its Cells Metabolism during Machine Perfusion"

_ijms, 2021, doi:10.3390/ijms22031121_

Round 1

Reviewer 1 Report

Assessing the quality of kidney grafts prior to transplantation is often challenging. In addition to the traditional lab results and medical history machine perfusion provides further opportunity for organ assessment. The authors reviewed the literature on the available assessment tools during machine perfusion.

In general, the paper is well written, however a linguistic review would be beneficial. Just to mention a few:

p2 line 100 'These are also the main responsible for...' 

line 108 "It is also to mention that. 

"cells" is used throughout the paper where organ would be more appropriate. (for example: lines 172, 188, 193)

The "Pathogenesis of the ischemic injury" chapter is detailed, gives the reader the background information on ischemic/reperfusion injury.

Compared to other chapters, this part is way too long. 

"Consequences of ischaemic reperfusion injury" - good summary on delayed graft function and acute rejection triggered by the reperfusion injury. 

Preservation techniques

Oxygenated machine perfusion is not mentioned in this chapter. Adding oxygen into the perfusion fluid has proven beneficial in several studies. Please include the COMPARE trial which has just been published in the Lancet.

Jochmans I, Brat A, Davies L et al. Oxygenated versus standard cold perfusion preservation in kidney transplantation (COMPARE): a randomised, double-blind, paired, phase 3 trial. The Lancet 2020; 396: 1653.

Normothermic Machine Perfusion

Normothermic perfusion can be divided in two supgroups: in situ (this is called Normothermic Regional Perfusion) and ex situ (when the individual organs are perfused separately after retreival, also referred to as 'ex vivo normothermic perfusion' or EVNP)

This is not mentioned in the manuscript, please include it!

Viability assessment

This is the main message of the paper, summarising the findings from 35 papers, but unfortunately this is also where the paper is significantly lagging behind. This chapter should be more detailed giving more information on the different assessment tools. Saying 'the quality assessment score (QAS), is based on macroscopic appearance, mean renal blood flow and total urine output [38].' is not enough, more information would be welcome. How does the score build up? What is the relevance for clinical practice? What did the researchers find with kidneys with lower and higher scores?

Similarly all 10 viability assessments should be more detailed.

Conclusion

This chapter is too succinct again, not summarising the findings, but trying to put the problem in a bigger perspective, more like an introduction.

In summary, I recommend a major review of the manuscript, detailing the findings of each study, focusing on possible clinical implementation. In addition a linguistic review would also be beneficial. 

Author Response

Reviewer1:

Assessing the quality of kidney grafts prior to transplantation is often challenging. In addition to the traditional lab results and medical history machine perfusion provides further opportunity for organ assessment. The authors reviewed the literature on the available assessment tools during machine perfusion.

We thank the reviewer for the time to assess our manuscript and for providing constructive criticism.

In general, the paper is well written, however a linguistic review would be beneficial. Just to mention a few:

p2 line 100 'These are also the main responsible for...' 

line 108 "It is also to mention that. 

We thank the reviewer for pointing this out. We have checked the manuscript and corrected grammatical mistakes and typos.

We have changed p2 line 100 as following: “When ROS production increases, oxidative damage follows in parallel”; line 108 has been changed in “furthermore”.

"cells" is used throughout the paper where organ would be more appropriate. (for example: lines 172, 188, 193)

Thank you; we have changed “cells” with “organ” and left the term “cell” when referring specifically to cell metabolism at mitochondrial level.

The "Pathogenesis of the ischemic injury" chapter is detailed, gives the reader the background information on ischemic/reperfusion injury.

Compared to other chapters, this part is way too long. 

Thank you, we thought that to provide a very detailed pathogenesis of the ischemic injury could have been beneficial to contextualise the machine perfusion tools in assessing it. We have expanded the other sections to better match with this one.

"Consequences of ischaemic reperfusion injury" - good summary on delayed graft function and acute rejection triggered by the reperfusion injury

Thank you.

Preservation techniques

Oxygenated machine perfusion is not mentioned in this chapter. Adding oxygen into the perfusion fluid has proven beneficial in several studies. Please include the COMPARE trial which has just been published in the Lancet.

Jochmans I, Brat A, Davies L et al. Oxygenated versus standard cold perfusion preservation in kidney transplantation (COMPARE): a randomised, double-blind, paired, phase 3 trial. The Lancet 2020; 396: 1653.

Thank you; we already mentioned this study, please kindly see p11 lines 292-294: “Recent studies have shown that oxygen administration during HMP increases oxygen consumption from the cells and improves kidney function (GFR) in the transplanted kidneys. We have also added following: In this regard, a recent randomised controlled trial has demonstrated that active oxygen addition into the perfusate, leads to a beneficial effect over HMP only in kidneys retrieved from donors aged 50 years or older or after circulatory death. The improvement shown in the 12-months eGFR in transplanted kidneys undergone oxygenation during HMP, paths the way towards real chances of reconditioning before transplantation, actively contrasting the ischemic damage as a result of the anaerobic cell metabolism.

We have also expanded the chapter adding the following paragraph, to mention the role of additional therapies during machine perfusion, in p5 lines 199-207: The aim of dynamic preservation with MP is to facilitate restoration of cellular metabolism, on the contrary impaired by SCS. Furthermore, there is the possibility to directly influence the perfusion fluid with the deliver of organ directed reconditioning therapies. In this unique scenario, by establishing an isolated ex‐vivo platform with a metabolically active organ, therapies targeting ischaemia reperfusion injury in real time could be delivered directly to the organ and limit systemic recipient exposure. In this regards, aside from the oxygen implementation, other drugs as prostaglandine, antibiotics, bicarbonate and heparin have been described, as well as mesenchymal cells in a recent pre-clinical trial.

Normothermic Machine Perfusion

Normothermic perfusion can be divided in two supgroups: in situ (this is called Normothermic Regional Perfusion) and ex situ (when the individual organs are perfused separately after retreival, also referred to as 'ex vivo normothermic perfusion' or EVNP)

This is not mentioned in the manuscript, please include it!

We thank the reviewer for pointing this out; however, the purpose of the present review is to assess organ viability during the ischemic storage outside the donor body, therefore, as suggested, we added the following: “Ex vivo NMP is to be distinguished from in situ normothermic regional perfusion, consisting of the use of extracorporeal membrane oxygenation in donors after circulatory death, while the organs are still within the donor body”.

Viability assessment

This is the main message of the paper, summarising the findings from 35 papers, but unfortunately this is also where the paper is significantly lagging behind. This chapter should be more detailed giving more information on the different assessment tools. Saying 'the quality assessment score (QAS), is based on macroscopic appearance, mean renal blood flow and total urine output [38].' is not enough, more information would be welcome. How does the score build up? What is the relevance for clinical practice? What did the researchers find with kidneys with lower and higher scores?

  • Thank you, we have significantly expanded this section with the incorporation of the review conducted (Table 1) into the main text and also providing more details for each single point of the viability assessment list. In particular for the QAS score we have added the following: Kidneys are graded 1–5, with 1-3 scores considered suitable for transplantation: score 1 indicates the least injury and 5 the most severe. More in details, the score is built up by a combination of the perfusion assessment parameters within 60 minutes from the start: grade I, excellent perfusion or global pink appearance; grade II, moderate perfusion with patchy pink/purple appearance which either remained or improves during NMP; grade III, poor perfusion, consisting of global mottling and purple/black appearance constantly throughout NMP. In addition, thresholds of renal blood flow (< 50 ml per min per 100 g) and total urine output (< 43 ml per min per 100 g) gives additional single points each to be combined with the macroscopic grades (I-III) for the final assessment score.

Similarly all 10 viability assessments should be more detailed.

Thank you, as this is a summary of the tools in view of the pathogenesis of the ischemic injury detailed above, we have expanded the section accordingly:

Point 2: Pressure, flow and resistance readings measured during MP are used as viability assessors, although they cannot be considered as stand-alone criteria, since their relative predictive value is low. The rationale for the use of perfusion parameters stands on the structure of the renal vascular system itself, very rich in capillary network with filtration function. The release of vasoconstrictors from this capillary network (single one-layer endothelium) following the ischemic and inflammatory insults, determines accumulation of erythrocytes and microthrombosis, eventually leading to a diminished flow and increased resistance in the graft. Furthermore, the hypoxia is directly responsible for endothelium cell activation, sinergically favouring a pro-coagulant and pro-inflammatory phenotype of the renal vasculature, with consequent disruption of the blood flow, and increased leukocyte infiltration, with a further decline in kidney function. On this basis, increased renal vascular resistance and low intraparenchymal flow are expressions of tissue damage.

Point 3 : Glucose consumption: the difference between the concentration into the arterial inflow and venous outflow could estimate the aerobic respiration and energy activity of the kidney cells. Several ways to measure glucose consumption have been described, including metabolic profiles via non invasive MR spectroscopy. The rationale lies on the estimation of cells viability in view of their metabolic utilization of carbohydrate energy sources, as it physiologically occurs when the organ is within the human body. The pattern of shutting down metabolically is peculiar of the kidney suffering from oxidative stress and shifting towards anaerobic energy production, while renal perfusion decreases.

Point 5 : Measurements of final glycolysis products. A lack of oxygen causes accumulation of peculiar metabolites: succinate/pyruvate, NADH, lactate (Figure 1). The measurement of tissue damage and estimated anaerobic metabolism is a major feature within ischaemic organs, with correlation to the extent of warm ischaemia time, as for example in the case of DCDs. 

Point 9 : mRNA profiling: defective postreperfusion metabolic recovery directly associates with incident delayed graft function and there is evidence of some ischemia induced omics that could be used as predictors of tissue injury. Specific mRNA expression of several glycolytic and gluconeogenic enzymes could evaluate renal glucose metabolism or the degree of inflammation and cytokine production, secondary to the ischemic insult .

Point 10 : Flavin mononucleotide (FMN) levels in the acellular perfusate after 30 minutes of hypothermic perfusion, as a result of damaged mitochondria releasing their content into cytoplasm. Physiologically, FMN is non-covalently bound to a subunit of the mitochondrial complex I and its dissociation with release at a cytoplasmatic level is an effect of the ischemic injury, where the MPT is damaged, with ROS production and increased toxicity. 

Conclusion

This chapter is too succinct again, not summarising the findings, but trying to put the problem in a bigger perspective, more like an introduction. 

We thank the reviewer, for highlighting this. The conclusion has been now modified as following: The development of dynamic organ perfusion technology has significantly increased the possibility to assess the parenchymal cells metabolism during preservation, thus offering an additional tool to consider organ viability, particularly for those organs retrieved from broader acceptance criteria, i.e. ECDs and DCDs. In this way, there is evidence of a concrete chance to expand the organ pool, through the variety of MP approaches, by tailoring each preservation parameter (temperature, oxygen and/or nutrient delivery, location) to each different organ on the basis of the presumed ischemic injury.

The concept of dynamic preservation itself is already beneficial to the organ metabolism, directly affecting the renal microvasculature and thus slowing down the resulting ischemic vasoconstrition. This could be estimated via the perfusion parameters of increased renal resistance and impaired intraparenchymal flow. Additionally, several indirect methods, as oxygen and glucose consumption, ATP depletion/production, lactate and biomarkers concentration could provide an insight of what is the current organ energy status to further drive reconditioning interventions. Yet, to date, there is a lack in the complete understanding of the mechanisms regulating the ischemic cell damage, thus a full viability assessment during dynamic organ perfusion is not available.

In summary, the more information related to viability assessment during organ preservation is provided, as in the case of MP compared to SCS, the highest is the accuracy in predicting the future function of a given graft. Further studies are warranted to reduce the inappropriate organ discard, possibly integrating a multidisciplinary approach to complement clinical or omics data with the variables examined via MP technology.

In summary, I recommend a major review of the manuscript, detailing the findings of each study, focusing on possible clinical implementation. In addition a linguistic review would also be beneficial. 

 Thank you

Reviewer 2 Report

With interest I read the manuscript by Bellini et al on the assessment of kidney graft viability and cell metabolism during machine perfusion. Machine perfusion is taking a flight in the field of organ transplantation, as it provides the opportunity to assess kidney function prior to transplantation. This manuscript nicely discusses the various biochemical pathways in the transplant kidney that may be assessed during the machine perfusion period to determine the quality of a transplant.

Major comments:

This manuscript starts with a detailed description of the pathogenesis and consequences of ischemic injury, followed by an overview of preservation techniques, which stand rather alone. These could be chapters of two separate papers. It would make the manuscript much stronger if both parts were intertwined. In paragraph 5 a point-wise sum up of potential measurements on kidneys on MP are given, but no details are given on how these parameters are expected to be affected by MP, or how they can influence clinical decision making. I suggest to delete paragraph 5 and incorporate (and discuss) all this information in the previous parts of the manuscript.

Other comments:

Page 4, line 129-133: Age is also an importance determinant for the reservoir of repair capacity of the kidney.

There are a several typos and grammatical errors, in particular in paragraph 4 and 5, which can be corrected easily with spelling control.

Paragraph 4 contains several very short subparagraphs, sometimes of only one sentence. This breaks up the flow of this part of the paper and reduces its readability. Please reorganize this part of the manuscript.

Some sentences in the manuscript do not read very well, for instance line 223, line 228, line 242, line 246. Please go through the manuscript to improve the grammar.

To state that normothermic machine perfusion maintains the organ as if it never left the body (line 235) is a too strong statement. The body hormonal regulatory system, interaction with other organs, day-night rhythm etc etc is missing in NMP.

Author Response

With interest I read the manuscript by Bellini et al on the assessment of kidney graft viability and cell metabolism during machine perfusion. Machine perfusion is taking a flight in the field of organ transplantation, as it provides the opportunity to assess kidney function prior to transplantation. This manuscript nicely discusses the various biochemical pathways in the transplant kidney that may be assessed during the machine perfusion period to determine the quality of a transplant.

We thank the reviewer for the time to assess our manuscript, the positive feedback and the constructive criticism.

Major comments:

This manuscript starts with a detailed description of the pathogenesis and consequences of ischemic injury, followed by an overview of preservation techniques, which stand rather alone. These could be chapters of two separate papers. It would make the manuscript much stronger if both parts were intertwined. In paragraph 5 a point-wise sum up of potential measurements on kidneys on MP are given, but no details are given on how these parameters are expected to be affected by MP, or how they can influence clinical decision making. I suggest to delete paragraph 5 and incorporate (and discuss) all this information in the previous parts of the manuscript.

We thank the reviewer for highlighting this; we have significantly expanded this section to better match the rest of the manuscript and discuss each point more in details; however, we preferred to leave the entire section as it was, because in this way we could better focus on single topics and left what we have already discussed within the other paragraphs.

Other comments:

Page 4, line 129-133: Age is also an importance determinant for the reservoir of repair capacity of the kidney.

Thank you, we agree that age is an important determinant, in fact we acknowledge this earlier on in the manuscript in the definition of ECD: age ≥ 60 years or over 50 years with ≥ 2 of the following conditions: hypertension, terminal serum creatinine equal or greater than 1,5 mg/dL or death resulting from an intracranial haemorrhage (page 1 lines 32-34). We have also added specifically the concept of aged donors as an important determinant for the recovery capacity, p.4 line 131: “---living donor versus ECD or DCD, or young versus aged donors”

There are a several typos and grammatical errors, in particular in paragraph 4 and 5, which can be corrected easily with spelling control.

Thank you, we have corrected them.

Paragraph 4 contains several very short subparagraphs, sometimes of only one sentence. This breaks up the flow of this part of the paper and reduces its readability. Please reorganize this part of the manuscript.

Thank you for pointing this out; we reorganised the section as suggested.

Some sentences in the manuscript do not read very well, for instance line 223, line 228, line 242, line 246. Please go through the manuscript to improve the grammar.

Thank you for pointing this out; we have gone through the manuscript as suggested.

To state that normothermic machine perfusion maintains the organ as if it never left the body (line 235) is a too strong statement. The body hormonal regulatory system, interaction with other organs, day-night rhythm etc etc is missing in NMP.

Thank you; we deleted this sentence.

Round 2

Reviewer 1 Report

Thank you for your reviewed manuscript.

The paper has improved significantly, easier to read. Several paragraphs have been rephrased and typos have been corrected.  

The viability assessment tools are presented in more detail and more useful for clinical practice.

Reviewer 2 Report

The authors have greatly improved their manuscript. It is now very well structured and includes useful discussion sections on various items which increase the importance of this paper for readers from the field.